# Nano-Organic Coatings Improve Early Vigor of *Brassica napus* L. Seeds in Water Deficit

Farkhondeh Abrahimi [1], Mansour Taghvaei [1,*] and Andrea Mastinu [2,*]

[1] Department of Plant Production and Genetics, School of Agriculture, Shiraz University, Shiraz 7144165186, Iran
[2] Department of Molecular and Translational Medicine, University of Brescia, Viale Europa 11, 25123 Brescia, Italy
* Correspondence: taghvaei@shirazu.ac.ir (M.T.); andrea.mastinu@unibs.it (A.M.)

**Abstract:** Low emergence and vigor of canola seeds are common problems encountered by canola growers. Seed coating is a good way to solve this problem. The objective of this study was to find the best combination of nano-organic to be used as a coating for seeds for strengthening the absorption of water and emergence seed of low vigor canola. The treatments in the first experiment included six levels of organic coatings: 20% vermicompost with 80% bentonite, 30% vermicompost with 70% bentonite, 40% vermicompost with 60% bentonite, 40% residues of canola with 60% bentonite, 30% residues of canola with 70% bentonite, and 20% resides of canola with 80% bentonite. The results indicated that 20% residues of canola and 20% vermicompost had the highest moisture under the saturation conditions and kept moisture for a longer time. Subsequently, the factors of the second experiment included three levels of water stress (100, 75, and 50% field capacity (FC)), three levels of seed coating (control, vermicompost, and canola residues), and three levels of titanium dioxide nanoparticles (0, 0.5, and 1 mM). The results showed that nano-organic coating improved emergence percentage and seedling traits while increasing superoxide dismutase, peroxidase, and catalase activities and decreasing electrolyte leakage. The highest ratios for emergence percentage (74.67), seedling vigor index (264.58), and leaf area (418) were obtained at the highest level of nano-titanium (1 mM). The highest rates of superoxide dismutase (53.44), peroxidase (30.68), catalase (5.35), chlorophyll content (51.05), and lowest electrolyte leakage (42.39) were observed in the highest level of nano-titanium (1 mM). Based on the results, we suggest the use of rapeseed residues with nano titanium oxide for rapeseed coating. The use of nanoparticles in current seed coatings is useful for reducing stresses in the germination and emergence stages.

**Keywords:** emergence rate; electrolyte leakage; superoxide dismutase activity; peroxidase activity

## 1. Introduction

As the first stage in plant production and development, seed germination plays an important role in the quality and quantity of final plant production [1–3]. The seeds have the ability to descend the genetic characteristics of the parental lines to the future generation; therefore, their own attributes, features, and genetic sources are very influential in crop production and final yield [1,2]. The seed vigor (early growth) can be affected by genetic characteristics and environmental factors such as light, temperature, humidity, etc.) [4–6]. Additionally, it has been reported that treating seeds with proper organic or inorganic compounds before planting, which is normally called pretreatment or priming, can improve the seed germination quality and quantity under different conditions compared to nontreated seeds [7,8]. Pretreatment of seeds is a simple and inexpensive but successful method to improve seed germination [9–11]. Other researchers claimed that seed pretreatment could also improve the germination and emergence percentage of seeds [12–14]. Different pretreatments of seeds with either organic or inorganic compounds have been implemented on different conditions and plant species [9,15,16]. Seed priming has been used to improve

germination, reduce seedling emergence time, and ameliorate stand establishment and yield [9]. Seed coating or seed pelleting is an important part of pretreatment [17,18]. In this method, changes in the shape, dry weight, and size of the seed or seed surface structure occur using different amounts of materials and compounds [19,20]. Seed coatings are used for different purposes, including protecting the seed from fungal and insect attack, protecting from birds and rodents, supplying micro- and macronutrients, supplying growth regulators, yielding more moisture absorption, supplying oxygen, germination stimulation, and germination delay [17,21,22]. On the other hand, plants are reportedly more sensitive to water stress at the germination stage [2,3,10,23–26]. Drought stress in this stage reduces the longitudinal growth of the shoot and root, delays the time of emergence, and reduces the uniformity of seedling emergence and density in the field [5,7,10,23]. In this regard, the coating method for pretreating seeds and improving their traits can bring about great advantages in this situation. The absorption of water, which depends on both water storage of the seed bed and seed characteristics, is the most essential factor for imbibition in the early stages of seed germination [27,28]. Lack of water in the seedbed leads to osmotic stress in seeds, potentially causing negative effects and preventing the seed from proper germination [29]. Coating seeds with proper materials can absorb higher amounts of water compared to nontreated seeds, resulting in the proper emergence and production of a stronger root system [18,19,30,31]. Some materials, such as plant residue, manure, and vermicompost, can maintain different amounts of moisture and improve seeds' ability to germinate more properly [32,33].

Materials that are used in seed coating typically consist of chalk, clay, soil, mica, quartz, talc, Pete, perlite, and wooden fiber, as well as all kinds of adhesives and varying levels of organic material [18]. Meanwhile, the application of nanotechnology and nanomaterials is progressively growing worldwide [34–36]. Nanoparticles are atomic or molecular complexes with dimensions as small as 1–100 nanometers with normally different properties compared to their original materials [37]. In this regard, the use of titanium dioxide nanoparticles has shown an increasing trend in plant production [11,29,35,36,38]. Titanium is one of the useful elements for plants that can stimulate the uptake of some nutrients such as nitrogen, phosphorus, calcium, magnesium, iron, manganese, and zinc depending on some biological factors, including plants species, pH, moisture, and nutrients in the soil [11,29,35,36,38]. Treated with titanium dioxide nanoparticles increase rates of germination, seedling dry weight, and vigor index of old soybean seeds compared to the control [36].

Canola (*Brassica napus*), an annual plant belonging to the Brassicaceae family, is an oil crop with a high yield potential and oil seed content in comparison to other oilseed plants [28,39]. Canola seed contains 40–45% oil and 25–35% protein. Canola oil has the best quality as an edible oil due to the perfect combination of unsaturated and low-saturated fatty acids. Canola seed is small and sensitive to drought stress during germination and seedling establishment [40]. Drought stress reduces the rate of daily germination and greatly reduces the number of established seedlings, and this problem occurs when replanting is not possible [40].

There are few reports on the use of nanoparticles in seed coating and almost no reports on the use of nano titanium dioxide in rape seed coating. Our hypothesis is that the use of organic coatings with titanium nanoparticles can reduce the drought stress in the germination and emergence stage by absorbing and retaining water around the rapeseed. This treatment should lead to an increase in the emergence rate and, ultimately, in the number of plants established. The novelty of this research is the combined use of organic matter, bentonite, and titanium nano-oxide materials to enhance the germination and emergence of rapeseed in the conditions of drought stress in the germination stage.

The aim of this research is to investigate the effect of using organic materials together with hydrophilic minerals such as bentonite and nano titanium oxide to create a hydrophilic compound capable of absorbing and retaining water for a longer period in the ground. The

combination of these components will create a coating suitable for improving germination and the emergence of oilseed rape under drought-stress conditions.

## 2. Materials and Methods

### 2.1. Experiment 1: Pattern of Water Absorption and Maintenance of Moisture by the Coated Seeds

Experiment 1 was performed to find the best combination of these treatments to be used as a coating for seeds to help seeds germinate more properly and, in turn, help seeds to emerge and absorb the moisture. The idea behind the experiment was that the materials used in the coating should be so sustainable that no harm or damage affects them during the planting in the field. They should also be able to respond to moisture properly so that the moisture can be easily absorbed. This experiment was carried out in a completely randomized design with five replications on canola seed var. Nepton. Treatments included six levels of organic coatings: C1—20% vermicompost with 80% bentonite; C2—30% vermicompost with 70% bentonite; C3—40% vermicompost with 60% bentonite; C4—40% residue of canola with 60% bentonite; C5—30% residues of canola with 70% bentonite; C6—20% residues of canola with 80% bentonite. Bentonite (sodium type) is a swelling clay consisting mostly of montmorillonite soil and is a powerful absorbent of water and liquids. Bentonite were purchased from Sinato Co (Shiraz, Iran). The residues of canola for this experiment were collected from the farms of the College of Agriculture, Shiraz University, Shiraz, Iran, while the vermicompost was prepared from Kimia Co. ltd. (Shiraz, Iran). The organic materials were dried in the shade at room temperature. They were then ground, powdered, and passed through a sieve with mesh 40. Bentonite, vermicompost (their characteristics are listed in Table 1), and residues were mixed at different rates with water to obtain a saturated dough-like form. The dough-like form of these mixtures was then placed in an incubator for 48 h at 30 °C and weighed at regular intervals to evaluate their ability to absorb and maintain moisture.

**Table 1.** Some chemical characteristics of vermicompost used.

| Vermicompost Properties | |
|---|---|
| pH | 7.75 |
| EC (ds/m) | 3.8 |
| Organic matter (%) | 44.2 |
| Organic carbon (%) | 25.6 |
| Total nitrogen (%) | 2.15 |
| Phosphorus (mg/kg) | 14,194 |
| Potassium (mg/kg) | 10,000 |
| Iron (mg/kg) | 3274 |
| Zinc (mg/kg) | 112.3 |
| Manganese (mg/kg) | 248.8 |
| Copper (mg/kg) | 28.7 |
| Canola residue Properties | |
| Cellulose (%) | 43 |
| Lignin (%) | 17 |
| Ash (%) | 6 |

### 2.2. Experiment 2: Effect of Moisture Regime, Organic Coating, and Titanium Dioxide Nanoparticles on Emergence Percentage, Emergence Rate, Seedling Vigor Index, Root and Shoot Dry Weight, and Leaf Area

After screening the treatments using the first experiment, Experiment 2 was conducted in 2019 in a greenhouse located in the College of Agriculture, Shiraz University, as a factorial experiment arranged in a completely randomized design with five replications. In this experiment, treatments included three levels of seed coating (SC0 = no coated seeds, SC1 = coated seeds with vermicompost, and SC2 = coated seeds with residues of canola), titanium dioxide nanoparticles at three levels (0 mM, 0.5 mM, and 1 mM), and water stress

at three moisture levels including well-watered (100% water holding capacity), moderate water stress (75% water holding capacity), and severe water stress (50% water holding capacity). Nanoparticles were purchased from Nano Pars Lima Co, and their characteristics are listed in Table 2.

**Table 2.** Some physical and chemical characteristics of titanium dioxide nanoparticles and soil used in the experiment pots.

| Texture Soil | | | | | | Properties | | | | | |
|---|---|---|---|---|---|---|---|---|---|---|---|
| **Sandy%** | **Clay%** | **Silt%** | **pH** | **EC (dS/m)** | **N%** | **P (mg/Kg)** | **K (mg/Kg)** | **Fe (mg/Kg)** | **Zn (mg/Kg)** | **Mn (mg/Kg)** | **Cu (mg/Kg)** |
| 42 | 20 | 38 | 7.54 | 0.72 | 0.2 | 16 | 266 | 2.2 | 2 | 3.2 | 0.4 |
| Nanoparticle | | | | Size (nM) | | | Purity (%) | | Surface (M$^2$/g) | | |
| Tio2 | | | | <30 | | | 99 | | <60 | | |

Before experiment initiation, the pots were filled with 5 kg of a mixture of soil, sand, and compost as 5:2:1. The soil texture was sandy loam, with the other characteristics of the soil presented in Table 2. In each pot, five seeds were sown. To apply water stress, the field capacity of the soil was determined, and the treatments were applied to the weighting base. Every alternate day, the pots were weighed, and water was added to achieve the target soil moisture.

*2.3. Data Collection*

The morphological traits, including emergence percentage, emergence rate, seedling vigor index, leaf area (Leaf Area Meter device using Delta Model-T Device), and seedling dry weight, (weighing through the samples dried at 70 °C for 24 h using scales with an accuracy of 0.001 g) were evaluated. Emergence percentage, mean time of emergence, and emergence rate were calculated by the following formulas [41].

$$EP = \text{Emerged seedling} \times 100/n$$

$$MTE = \Sigma TiNi/\Sigma Ni$$

$$ER = 1/MTE$$

where EP represents the emergence percentage, n is the number of seeds planted in each plot, MTE denotes the mean time of emergence, Ti refers to the number of seeds emerging per day, Ni represents the total seed emergence on each measurement day, and ER shows the emergence rate.

The seedling vigor index was calculated using the following formula:

$$VI = EP \times SL$$

where VI is the seedling vigor index, and SL is the seedling length.

The activity of superoxide dismutase, peroxidase, catalase enzymes, and electrolyte leakage was calculated by Beauchamp and Fridovich [41], Chance and Maehly [42], and Dhindsa [43], respectively.

*2.4. Statistical Analysis*

Prior to analysis, the data population normality was verified by Kolmogorov–Smirnov with SPSS (10.0) software (IBM, Armonk, New York, NY, USA). The data were analyzed using 'SAS' statistical software v.9.3 (SAS Institute Inc., Cary, NC, USA) after ensuring the homogeneity of variances. The means were compared by Duncan's multiple range test (DMRT) in cases where the F-test of the ANOVA table indicated a significant difference, at

least at the $p < 0.05$ and $p < 0.01$ level, and the curve was drowned by Excel (Microsoft Inc., Chicago, IL, USA). Figure 1 shows the experimental design.

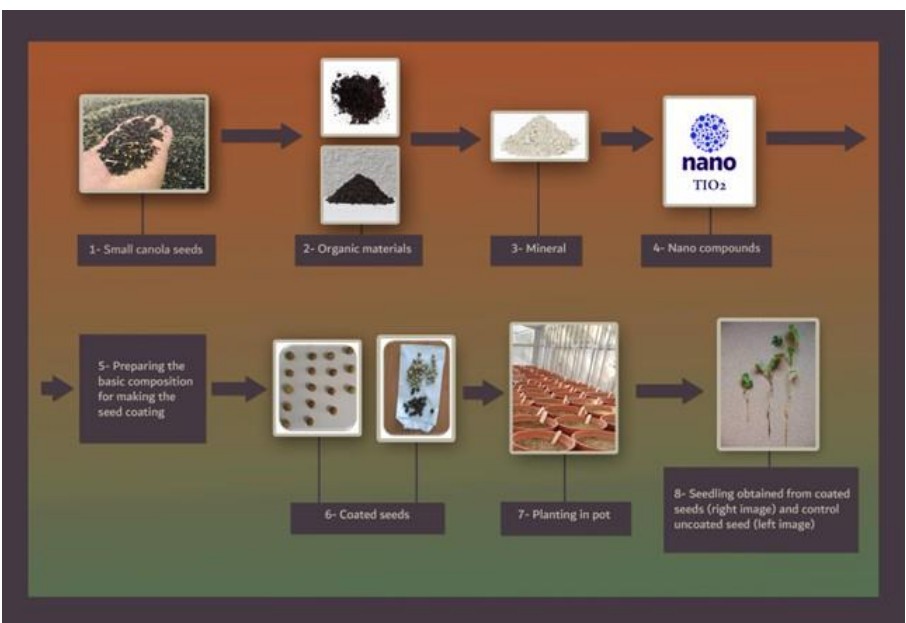

**Figure 1.** The figure shows the experimental protocol followed.

### 3. Results

*3.1. Experiment 1: Pattern of Water Absorption and Maintenance of Moisture by the Organic Matter for the Seed Coating*

Organic matter and time after saturation had significant effects on water absorption and maintenance of moisture by the coated seeds. The results of the mean comparison for water absorption showed that 20% vermicompost with 80% bentonite and 20% canola residue with 80% bentonite had the highest water absorption under saturation conditions (Figure 2) with no significant difference. The slopes of the regression lines for these treatments represented that 20% application of either canola residue or compost mixed with bentonite yielded the highest constant and the lowest negative trend for loss of the absorbed moisture (Table 3). Moreover, the average saturation content for organic treatments indicated that after taking the coated seeds out of the water, the saturation content was the highest when applying the residue and compost at 20% for each without any significant differences. Since the use of 20% of either compost or canola residue resulted in the highest saturation content followed by the lowest rate of moisture loss over time, these two treatments have been the best organic proportions for usage as a seed coating for canola seeds compared to the other treatments in this experiment (Table 3).

**Table 3.** Linear model of weight in saturation condition and moisture content in kinds of organic coating.

| Kind of Coating | Average Saturation | Content of Maintain Moisture Model | $R^2$ | — |
|---|---|---|---|---|
| 20% vermicompost | 3.219 [a] | Y = −0.124x + 3.417 | 0.981 | −0.124 |
| 30%vermicompost | 2.979 [b] | Y = −0.168x + 3.207 | 0.984 | −0.168 |
| 40% vermicompost | 2.914 [b] | Y = −0.170x + 3.092 | 0.978 | −0.170 |
| 20% canola residue | 3.305 [a] | Y = −0.132x + 3.433 | 0.984 | −0.132 |
| 30% canola residue | 3.032 [ab] | Y = −0.157x + 3.256 | 0.963 | −0.157 |
| 40% canola residue | 2.821 [b] | Y = −0.163x + 3.372 | 0.961 | −0.163 |

Significant differences ($p < 0.05$ or $p < 0.01$) were indicated with different letters.

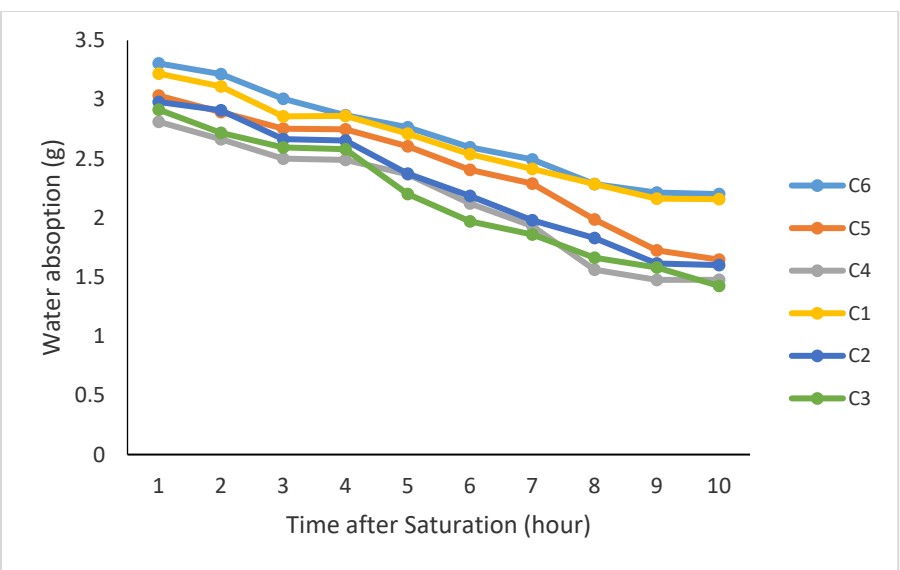

**Figure 2.** Content of water absorbed by the coated seeds under different coating treatments after saturation in water (C1: 20% vermicompost with 80% bentonite, C2: 30% vermicompost with 70% bentonite, C3: 40% vermicompost with 60% bentonite, C4: 40% rapeseed residues with 60% bentonite, C5: 30% rapeseed residues with 70% bentonite, and C6: 20% rapeseed residues with 80% bentonite).

### 3.2. Experiment 2: Effect of Moisture Regime, Organic Coating, and Titanium Dioxide Nanoparticles on Emergence Percentage, Emergence Rate, Seedling Vigor Index, Shoot and Root Dry Weight, and Leaf Area

In the second experiment, different proportions of titanium dioxide nanoparticles were added to the selected proportion of the coating material (20% canola residue and 20% vermicompost mixed with bentonite) from the first experiment to measure their ability to induce tolerance to water stress conditions. Water stress had a significant effect on emergence percentage, emergence rate, seedling vigor index, root and shoot dry weight, and leaf area (Table 4). Water stress significantly decreased the emergence percentage and seedling traits (Table 5). Emergence percentage, emergence rate, seedling vigor index, root and shoot dry weight, and leaf area decreased with increasing water stress (Table 5). The highest emergence percentage (77.33), emergence rate (0.21), seedling vigor index (302.22), shoot dry weight (7.31 g), root dry weight (17.1 g), and leaf area (500 cm$^2$) were obtained under 100% FC condition, while the lowest rate for all of these traits was observed in under 50% FC condition (Table 5). Coating with the organic matter had a significant effect on emergence percentage, emergence rate, and seedling traits (Table 4). The highest emergence percentage (83.56), emergence rate (0.22), seedling vigor index (237.33), shoot dry weight (5.67 g), root dry weight (14.02 g), and leaf area (471) were obtained in the coating with canola residue, while the lowest rate for all these traits was observed in the control with no seed coating (Table 5).

**Table 4.** Analysis of variance for emergence, emergence rate, seedling vigor index, shoot dry weight, root dry weight, and leaf area as affected by moisture regime, coating, and nanoparticle.

| | | Mean Square | | | | | |
|---|---|---|---|---|---|---|---|
| **Source** | **df** | **Emergence (%)** | **Emergence Rate (Seed/Day)** | **Seedling Vigor Index** | **Shoot Dry Weight (g)** | **Root Dry Weight (g)** | **Leaf Area (g)** |
| Moisture regime(W) | 2 | 3567.4 ** | 0.06 ** | 226,439.23 ** | 79.22 ** | 0.99 ** | 307,868.7 ** |
| Coating (C) | 2 | 12,625.18 ** | 0.07 ** | 227,841.98 ** | 44.53 ** | 41.07 ** | 172,962.42 ** |
| Nanoparticle(N) | 2 | 2809.96 ** | 0.06 ** | 49,176.89 ** | 80.36 ** | 504.23 ** | 284,450.94 ** |
| W*C | 4 | 2.96 ns | 0.001 ** | 5559.10 * | 0.00 ns | 0.00 ns | 0.00 ns |
| W*N | 4 | 38.75 ns | 0.001 * | 683.11 * | 0.98 ns | 4.23 ns | 1550.48 ns |
| C*N | 4 | 65.97 ns | 0.00 ns | 504.57 ns | 0.42 ns | 17.21 ns | 3269.54 ns |
| W*C*N | 4 | 30.157 ns | 0.00 * | 443.86 ns | 0.87 ns | 2.46 ns | 1326.31 ns |
| Error | 104 | 245.20 | 0.00 | 2942.99 | 0.54 | 5.89 | 2846.09 |
| C.V% | | 23.08 | 8.56 | 22.82 | 12.86 | 18.40 | 12.79 |

**, * and ns: significant at the 1%, 5% probability level and not significant, W = Moisture regime, C = Coating, N = Nanoparticle.

**Table 5.** Effect of moisture regime, organic coating, and titanium dioxide nanoparticles on emergence traits.

| **Treatment** | **Emergence Percentage** | **Emergence Rate** | **Seedling Vigor Index** | **Shoot Dry Weight (g)** | **Root Dry Weight (g)** | **Leaf Area (cm$^2$)** |
|---|---|---|---|---|---|---|
| Moisture regime (%FC) | | | | | | |
| 100 | 77.33 [a] | 0.21 [a] | 302.22 [a] | 7.31 [a] | 17.16 [a] | 500 [a] |
| 75 | 69.33 [b] | 0.17 [b] | 248.80 [b] | 5.62 [b] | 12.81 [b] | 431 [b] |
| 50 | 58.22 [c] | 0.14 [c] | 161.69 [c] | 4.28 [c] | 9.56 [c] | 322 [c] |
| Coating | | | | | | |
| Control | 51.56 [c] | 0.14 [c] | 166.53 [c] | 2.97 [c] | 10.65 [c] | 238 [c] |
| Vermicompost | 69.78 [b] | 0.17 [b] | 237.33 [b] | 5.67 [b] | 13.20 [b] | 424 [b] |
| Residue | 83.56 [a] | 0.22 [a] | 308.84 [a] | 6.72 [a] | 14.02 [a] | 471 [a] |
| Nano titanium dioxide (mM) | | | | | | |
| control | 60.00 [c] | 0.14 [b] | 202.00 [c] | 4.70 [b] | 12.20 [b] | 353 [b] |
| 0.5 | 70.22 [b] | 0.18 [a] | 246.13 [b] | 5.99 [a] | 13.46 [a] | 435 [a] |
| 1 | 74.67 [a] | 0.21 [a] | 264.58 [a] | 5.78 [a] | 13.20 [a] | 418 [a] |

Data representation of three repeated experiments. Means followed by the same letter in a column were not different by Duncan's multiple range test ($p \leq 0.05$).

The emergence percentage, emergence rate, seedling vigor index, shoot dry weight, root dry weight, and leaf area significantly affected by titanium nanoparticles (Table 4). The application of titanium nanoparticles improved the emergence and seedling traits of seedlings (Table 5). The highest ratios for emergence percentage (74.67), emergence rate (0.21), seedling vigor index (264.58), shoot dry weight (5.78), root dry weight (13.20), and leaf area (418) were obtained at the highest level of nano-titanium (1 mM), while their lowest rates were observed in the control with no application of this nanomaterial (Table 5).

The interaction of coating with organic matter and water stress had a significant effect on the emergence rate and seedling vigor index (Table 6). The emergence rate of canola seeds increased from 0.18 to 0.27(seed/day) by applying the canola residue under 100% FC and just from 0.11% to 0.18% under 50% FC. Similarly, the seedling vigor index increased from 217 under control conditions under no water stress to 359 in canola residue application, while these ratios under 50% FC were from 139 to 263, respectively (Table 6).

**Table 6.** Effect of moisture regime and organic coating on emergence rate and seedling vigor index.

| Moisture Regime | 100% FC | | | 75% FC | | | 50% FC | | |
|---|---|---|---|---|---|---|---|---|---|
| Coating | $C_0$ | $C_1$ | $C_2$ | $C_0$ | $C_1$ | $C_2$ | $C_0$ | $C_1$ | $C_2$ |
| Emergence Rate | 0.18 cd | 0.20 c | 0.27 a | 0.15 d | 0.17 cd | 0.21 b | 0.11 e | 0.14 d | 0.18 cd |
| Seedling Vigor Index | 217 c | 294 b | 359 a | 184 cd | 256 bc | 302 ab | 139 e | 213 c | 263 bc |

Data represented from three replication. Means followed by the same letter in each row were not different by Duncan's multiple range test ($p \leq 0.05$), FC = field capacity, $C_0$ = no coated seeds, $C_1$ = coated seeds with vermicompost, and $C_2$ = coated seeds with residues of canola.

The emergence rate and seedling vigor index were significantly affected by the interaction of nanoparticles and water stress (Table 4). The emergence rate of canola seeds increased from 0.17 to 0.25 (seed/day) by applying the nanoparticles under 100% FC and only from 0.11% to 0.17% under 50% FC (Table 7). Similarly, the seedling vigor index increased from 248 under control conditions under no water stress to 352 by applying the nanoparticles under 100% FC, while these ratios under 50% FC were only from 175 to 255, respectively (Table 7).

**Table 7.** Effect of moisture regime and nano titanium dioxide on emergence rate and seedling vigor index.

| Moisture Regime | 100% F.C | | | 75% F.C | | | 50% F.C | | |
|---|---|---|---|---|---|---|---|---|---|
| Nanoparticle | Control | 0.5 mM | 1 mM | Control | 0.5 mM | 1 mM | Control | 0.5 mM | 1 mM |
| Emergence Rate | 0.17 b | 0.22 ab | 0.25 a | 0.14 bc | 0.18 b | 0.21 ab | 0.11 c | 0.15 bc | 0.17 b |
| Seedling vigor Index | 248 bc | 337 a | 352 a | 220 c | 279 b | 295 b | 175 d | 243 bc | 255 bc |

Data represented from three replication. Means followed by the same letter in each row were not different by Duncan's multiple range test ($p \leq 0.05$), FC = field capacity, and mM = millimolar.

Water stress had a significant effect on superoxide dismutase, peroxidase, catalase, electrolyte leakage, and chlorophyll content (Table 8). Superoxide dismutase, peroxidase, catalase, and electrolyte leakage increased with increasing water stress, but chlorophyll content decreased with increasing water stress (Table 9). The lowest superoxide dismutase (17.34), peroxidase (18.71), catalase (1.77), and electrolyte leakage (29.438) were obtained under 100% FC condition, while the highest rate for all these traits was observed in under 50% FC condition (Table 9). The highest chlorophyll content (54.22) was obtained under 100% FC condition, while the lowest rate of it was observed in under 50% FC condition (Table 9). Coating with an organic matter had a significant effect on superoxide dismutase, peroxidase, catalase, and electrolyte leakage (Table 8). The lowest superoxide dismutase (27.04), peroxidase (20.88), catalase (3.69), and chlorophyll content (48.31) and the highest electrolyte leakage (52.17) were obtained in the no-seed coating. On the other hand, the highest rate of superoxide dismutase, peroxidase, catalase, and chlorophyll content and the lowest electrolyte leakage were observed in the coating with canola residue (Table 9). The superoxide dismutase, peroxidase, catalase, and electrolyte leakage were significantly affected by titanium nanoparticles (Table 8). The lowest ratios for superoxide dismutase (34), peroxidase (23.49), catalase (3.71), and chlorophyll content (48.60) and the highest electrolyte leakage (50.04) were obtained at the control with no application of this nanmaterial. However, their highest rates of superoxide dismutase, peroxidase, catalase, and chlorophyll content and lowest electrolyte leakage were observed in the highest level of nano-titanium (1 mM) (Table 9). The interaction of water stress and coating with an organic matter had a significant effect on superoxide dismutase, peroxidase, catalase, electrolyte leakage, and chlorophyll content (Table 8). The superoxide dismutase increased only from 14.57 (in the no-seed coating) to 20.63 by applying the canola residue under 100% FC but from 45.13 (in the no-seed coating) to 79.07 under 50% FC (Table 10). The peroxidase increased just from 10.25 (in the no-seed coating) to 24.24 by applying the canola

residue under 100% FC but from 31.59 (in the no-seed coating) to 53.64 under 50% FC (Table 10). The catalase increased only from 1.40 (in the no-seed coating) to 2.25 by applying the canola residue under 100% FC but from 5.59 (in the no-seed coating) to 8.55 under 50% FC (Table 10). The electrolyte leakage decreased just from 33.32 (in the no-seed coating) to 26.35 by applying the canola residue under 100% FC but from 74.20 (in the no-seed coating) to 55.13 under 50% FC (Table 10). The interaction of water stress and nanoparticle had a significant effect on superoxide dismutase, peroxidase, and catalase (Table 8). The superoxide dismutase increased just from 12.70 (no application of nanoparticle) to 33.50 by applying the nanoparticle under 100% FC but from 66.03 (no application of nanoparticle) to 73.07 under 50% FC (Table 11). The peroxidase increased just from 15.58 (no application of nanoparticle) to 21.28 applying the nanoparticle under 100% FC but from 34.05 (no application of nanoparticle) to 44.77 under 50% FC (Table 11). The catalase increased only from 1.35 (no application of nanoparticle) to 2.11 by applying the nanoparticle under 100% FC but from 5.67 (no application of nanoparticle) to 8.23 under 50% FC (Table 11). The interaction of coating with organic matter and nanoparticles had a significant effect on superoxide dismutase and catalase (Table 8). The superoxide dismutase increased only from 24.17 (no application of nanoparticle) to 28.80 by applying the nanoparticle in the no-seed coating but from 45.23 (no application of nanoparticle) to 66.17 in seed coated with canola residue (Table 12). The catalase increased just from 3.45 (no application of nanoparticle) to 4.14 by applying the nanoparticle in the no-seed coating, but from 4.28 (no application of nanoparticle) to 6.35 in seed coated with canola residue (Table 12).

**Table 8.** Analysis of variance for superoxide dismutase, peroxidase, catalase, electrolyte leakage, and chlorophyll content as affected by moisture regime, coating, and nanoparticle.

| | | **Mean Square** | | | | |
|---|---|---|---|---|---|---|
| **Source** | **df** | **Superoxide DISMUTASE (u mg$^{-1}$ Protein)** | **Peroxidase (u mg$^{-1}$ Protein)** | **Catalase (u mg$^{-1}$ Protein)** | **Electrolyte Leakage (%)** | **Chlorophyll Content (mgg1fw)** |
| Moisture regime(W) | 2 | 40,856.80 ** | 27.9 ** | 346.25 ** | 12,392.32 ** | 660.919 ** |
| Coating (C) | 2 | 21,832.46 ** | 5477.28 ** | 36.60 ** | 1775.12 ** | 215.92 ** |
| Nanoparticle(N) | 2 | 5124.29 ** | 2169.89 ** | 34.20 ** | 841.81 ** | 111.17 ** |
| W*C | 4 | 3666.24 ** | 468.74 ** | 6.0 ** | 165.35 ** | 14.03 ** |
| W*N | 4 | 485.50 ns | 37.57 * | 4.07 ** | 30.83 ns | 6.05 ns |
| C*N | 4 | 1198.74 ** | 4.89 ns | 0.70 ** | 0.24 ns | 4.07 ns |
| W*C*N | 4 | 705.19 ** | 21.38 * | 0.70 ** | 53.27 ** | 0.64 ns |
| Error | 104 | 207.97 | 10.18 | 0.019 | 13.92 | 4.87 |
| C.V% | | 16.63 | 11.57 | 9.17 | 8.28 | 4.38 |

**, * and ns: significant at the 1%, 5% probability level and not significant. W = Moisture regime, C = Coating, and N = Nanoparticle.

**Table 9.** Effect of moisture regime, organic coating, and nano titanium dioxide on enzymatic antioxidant, electrolyte leakage, and chlorophyll content.

| | **100% FC** | **75% FC** | **50% FC** |
|---|---|---|---|
| | | Moisture regime | |
| Superoxide dismutase (u mg$^{-1}$ protein) | 17.34 [c] | 55.46 [b] | 67.57 [a] |
| Peroxidase (u mg$^{-1}$ protein) | 18.71 [c] | 24.08 [b] | 39.93 [a] |
| Catalase (u mg$^{-1}$ protein) | 1.77 [c] | 5.04 [b] | 7.29 [a] |
| Electrolyte leakage (%) | 29.43 [a] | 43.31 [b] | 62.48 [c] |
| chlorophyll content (mgg1FW) | 54.22 [a] | 50.14 [b] | 46.56 [c] |
| | $C_0$ | $C_1$ | $C_2$ |

**Table 9.** *Cont.*

|  | 100% FC | 75% FC | 50% FC |
|---|---|---|---|
|  | Organic coating | | |
| Superoxide dismutase (u mg$^{-1}$) | 27.04 [b] | 54.33 [a] | 58.99 [a] |
| Peroxidase (u mg$^{-1}$ protein) | 20.88 [c] | 27.10 [b] | 34.74 [a] |
| Catalase (u mg$^{-1}$ protein) | 3.69 [c] | 5.00 [b] | 5.42 [a] |
| Electrolyte leakage (%) | 52.17 [c] | 43.48 [b] | 39.56 [a] |
| chlorophyll content (mgg1FW) | 48.31 [c] | 49.93 [b] | 52.68 [a] |
|  | Control | 0.5 Mm | 1 Mm |
|  | Nanoparticle | | |
| Superoxide dismutase (u mg$^{-1}$) | 34.00 [b] | 52.92 [a] | 53.44 [a] |
| Peroxidase (u mg$^{-1}$ protein) | 23.49 [c] | 28.55 [b] | 30.68 [a] |
| Catalase (u mg$^{-1}$ protein) | 3.71 [b] | 5.04 [a] | 5.35 [a] |
| Electrolyte leakage (%) | 50.04 [b] | 42.78 [a] | 42.39 [a] |
| chlorophyll content (mgg1FW) | 48.60 [b] | 51.28 [a] | 51.05 [a] |

Data represented from three replication. Means followed by the same letter in each row were not different by Duncan's multiple range test ($p \leq 0.05$), $C_0$ = no coated seeds, $C_1$ = coated seeds with vermicompost, $C_2$ = coated seeds with residues of canola, and Mm= millimolar.

**Table 10.** Effect of moisture regime and organic coating on enzymatic antioxidant, electrolyte leakage, and chlorophyll content.

| Moisture Regime | 100% FC | | | 75% FC | | | 50% FC | | |
|---|---|---|---|---|---|---|---|---|---|
| Coating | $C_0$ | $C_1$ | $C_2$ | $C_0$ | $C_1$ | $C_2$ | $C_0$ | $C_1$ | $C_2$ |
| Superoxide dismutase (u mg$^{-1}$) | 14.57 [e] | 16.83 [d] | 20.63 [d] | 21.43 [d] | 67.67 [b] | 77.27 [a] | 45.13 [c] | 78.50 [a] | 79.07 [a] |
| Peroxidase (u mg$^{-1}$ protein) | 10.25 [d] | 21.64 [cd] | 24.24 [c] | 20.79 [cd] | 25.11 [bc] | 26.34 [bc] | 31.59 [bc] | 34.56 [ab] | 53.64 [a] |
| Catalase (u mg$^{-1}$ protein) | 1.40 [f] | 1.66 [f] | 2.25 [e] | 4.07 [d] | 5.60 [c] | 5.44 [cd] | 5.59 [c] | 7.72 [b] | 8.55 [a] |
| Electrolyte leakage (%) | 33.32 [b] | 28.61 [a] | 26.35 [a] | 49.60 [d] | 43.12 [c] | 37.21 [bc] | 74.20 [f] | 58.70 [e] | 55.13 [e] |
| Chlorophyll content (mgg1FW) | 44.56 [e] | 45.80 [e] | 49.31 [d] | 48.69 [d] | 50.23 [c] | 51.51 [c] | 51.67 [c] | 53.77 [b] | 57.22 [a] |

Data represented from three replication. Means followed by the same letter in each row were not different by Duncan's multiple range test ($p \leq 0.05$), C0 = no coated seeds, C1 = coated seeds with vermicompost, and C2 = coated seeds with residues of canola. FC = field capacity.

**Table 11.** Effect of moisture regime and nanoparticle on superoxide dismutase, peroxidase, and catalase.

| Moisture Regime | 100% FC | | | 75% FC | | | 50% FC | | |
|---|---|---|---|---|---|---|---|---|---|
| Nanoparticle | Control | 0.5 mM | 1 mM | Control | 0.5 mM | 1 mM | Control | 0.5 mM | 1 mM |
| Superoxide dismutase (u mg$^{-1}$) | 12.70 [f] | 19.26 [e] | 20.07 [e] | 33.50 [d] | 66.03 [b] | 66.43 [b] | 55.80 [c] | 73.07 [a] | 73.83 [a] |
| Peroxidase (u mg$^{-1}$ protein) | 15.58 [e] | 19.28 [d] | 21.28 [cd] | 20.86 [cd] | 25.38 [c] | 25.99 [c] | 34.05 [b] | 40.98 [ab] | 44.77 [a] |
| Catalase (u mg$^{-1}$ protein) | 1.35 [f] | 1.85 [ef] | 2.11 [e] | 4.12 [d] | 5.30 [cd] | 5.70 [c] | 5.67 [c] | 7.97 [ab] | 8.23 [a] |

Data represented from three replication. Means followed by the same letter in each row were not different by Duncan's multiple range test ($p \leq 0.05$), FC = field capacity, mM = millimolar.

**Table 12.** Effect of organic coating and nanoparticle on superoxide dismutase and catalase.

| Organic Coating | C₀ | | | C₁ | | | C₂ | | |
|---|---|---|---|---|---|---|---|---|---|
| Nanoparticle | Control | 0.5 mM | 1 mM | Control | 0.5 mM | 1 mM | Control | 0.5 mM | 1 mM |
| Superoxide dismutase (u mg$^{-1}$) | 24.17 [de] | 28.17 [d] | 28.80 [d] | 32.60 [c] | 65.03 [a] | 65.37 [a] | 45.23 [b] | 65.57 [a] | 66.17 [a] |
| Catalase (u mg$^{-1}$ protein) | 3.45 [d] | 4.06 [c] | 14.4 [c] | 3.98 [cd] | 5.45 [b] | 5.56 [ab] | 4.28 [c] | 5.61 [ab] | 6.35 [a] |

Data represented from three replication. Means followed by the same letter in each row were not different by Duncan's multiple range test ($p \leq 0.05$), C0 = no coated seeds, C1 = coated seeds with vermicompost, and C2 = coated seeds with residues of canola, mM= millimolar.

## 4. Discussion

Drought stress is one of the main causes of the depletion of plant establishment in arid and semi-arid regions [44,45]. Water uptake of seeds is a basic requirement for germination and development of seedling emergence [46]. Low rainfall, high evaporation, and lack of organic materials in the soil (<1%) due to sparse coverage and negligible vegetation are characteristics of dryland areas [47]. Water stress due to low absorption and water durability in the soil is one of the reasons for reduced germination and plant establishment in dry land areas. It is very difficult to increase organic matter in all soil masses for water absorption and long-term maintenance under water stress [48]. Nevertheless, with the use of organic seed coatings, a proper place can be created for the early growth of seeds. The use of organic matter creates a safe site thanks to improving absorption and prolonging the maintenance of water [31]. Adding organic matter to the soil can increase the amount of water absorption in the saturation stage (Figure 2). The maintenance of water absorbed by soil is also one of the problems of dry land. The use of organic materials or adding them to the soil reduces the slope of water loss (Table 5). Water uptake of seeds is a basic requirement for germination and development of seedling emergence. Water uptake by seeds is delayed under water stress conditions. Organic materials create positive changes in the seed surroundings by mitigating the effects of water stress and making nutrient supplies more available for planted crops during the growing season. They also provide optimal conditions for their growth and enhance the emergence percentage and emergence rate [17–19,21,30,31]. The use of 20% vermicompost with 80% bentonite and 20% canola residue with 80% bentonite significantly improved the emergence percentage and emergence rate (Table 6). Similar results have been reported with the use of vermicompost, the positive effect of vermicompost on reducing the effect of water deficit on germination and supplying plant hormones [49,50].

Nanomaterials can enhance the ability of seeds to absorb and use water from the soil. They also improve the emergence and seedling traits under water stress. The absorption of higher moisture from the media and surrounding soil of the seedling is normally possible using organic materials as a seed coating. However, this influence is more significant under water stress conditions because of the limited supply of water in this situation. The combination of titanium dioxide nanoparticles and selenium dioxide nanoparticles accelerates seed germination and seedling growth at a notable level [11,29,35,36,38]. In other research, titanium dioxide nanoparticles increased the lentil seedling length [11]. Nanoparticles increased the root growth of barley (*Hordeum vulgare* L.) in comparison to the control [38,51]. Titanium dioxide nanoparticles increased the germination rate of spinach (*Spinacia oleracea*) [52]. Similar results have been reported with the use of nanoparticles in seed coatings [53]. According to them, the reason is that nanoparticles have great permeability and quickly penetrate the seed coating, leading to the uptake of water under water stress, culminating in improved germination properties and seedling emergence. The application of titanium nanoparticles improved the emergence percentage, emergence rate, seedling vigor index, shoot dry weight, root dry weight, and leaf area under water stress (Table 7). Similar results have been reported about the positive effects of nanoparticles

on *Zea mays* early vigor [36]. The application of nano-titanium in the organic mixture of seed coats could enhance the activities of the antioxidant system, helping the plants cope with the oxidants produced under water stress conditions and grow more properly (Table 7). When plants are exposed to environmental stresses, a variety of reactive oxygen species (ROS), such as superoxide anion radical, hydrogen peroxide, hydroxyl radicals, and hydrogen peroxide, induce oxidative stress within the cells, which may ultimately lead to cell death [54]. In response to this condition, the cells normally generate some sort of antioxidant defense, such as superoxide dismutase, peroxidase, and catalase [23,36]. In the current study, it was observed that the application of both organic matter (canola residue and compost) and titanium nanoparticles could enhance the efficiency of the antioxidant defenses in canola seedlings. Nanoparticles can change the intracellular conditions of cells during their entrance, enabling them to absorb higher levels of elements required for enzymatic and hormonal productions, such as iron, zinc, and magnesium [55]. The higher the availability of micronutrients in cells, the greater the suitability of the cells to produce hormones and enzymes to respond to the condition. Indeed, by enhancing the efficiency of the enzymatic system against the produced oxidant, the application of organic matter and nanoparticles in seed coating proved to be a suitable and practical method to cope with oxidative stress under water shortage conditions [18,19]. Further, electrolyte leakage is an index for determining the damages induced to the cell walls [56]. On the other hand, its content decreased with the application of both the seed coating and nano-titanium. Organic seed coating and titanium nanoparticles effectively reduced its content compared to no application of these materials (Table 12). One of the most important negative effects of ROS is invading the cell walls and damaging them [54]. By enhancing the antioxidant efficiency of the cells, organic coating and nanoparticles encouraged a strong tolerance against the oxidative stress caused by water stress in canola. Water stress also reduces leaf chlorophyll content and subsequently compromises the seedling growth of leaf chlorophyll (Table 10). Similar results regarding the reduction in chlorophyll following drought stress have been reported by other researchers [23,26,39,57].

The use of organic material as seed coating can reduce the negative effects of water stress. According to similar results, organic matter application as seed coating can improve the chlorophyll content and photosynthesis rate in rice plants [20]. Environmental stress decreases the chlorophyll content with more production of oxygen radicals in the cell, eventually causing oxidation and degradation of pigments [58].

## 5. Conclusions

The overall results of the first experiment revealed that the seed coating with organic matter enhanced water retention around the seed. The best combination rate of canola residue and vermicompost as a seed coating for canola was either 20% of canola residual or 20% of vermicompost in a mixture of 80% bentonite, yielding the highest performance of the seedling. The results of the second experiment suggested that the application of either canola residue or vermicompost combination with titanium can improve seedling-related traits, such as emergence percentage, seedling vigor index, emergence rate, and root and shoot dry weight by increasing water uptake under water stress conditions. The application of titanium nanoparticles as an admixture of seed coating with two other organic compounds resulted in higher efficiency in responding to the negative impacts of water stress. Further, water stress enhances the activities of antioxidant enzymes, including superoxide dismutase, peroxidase, and catalase. However, the application of either nanoparticles or organic materials as seed coating showed to increase their activities under both drought stress and normal conditions, where their activities reached a higher level under water stress in comparison to no application of these materials. Electrolyte leakage and chlorophyll content of the canola seedlings negatively responded to water stress while positively responding to organic materials and nanoparticles. The final suggestion is that rapeseed seeds should be covered using a combination of organic materials, bentonite, and titanium nanoparticles for planting in dry ecosystems so that they are able to cope with

the negative effects of water stress and produce more emergence in these conditions. This method is recommended for commercial canola seed sellers.

**Author Contributions:** Conceptualization, M.T. and F.A.; methodology, M.T. and F.A.; formal analysis, M.T. and F.A.; data curation, M.T., F.A. and A.M.; writing—original draft preparation, M.T. and F.A; writing—review and editing, M.T. and A.M.; supervision, M.T.; project administration, M.T. and A.M.; funding acquisition, A.M. All authors have read and agreed to the published version of the manuscript.

**Funding:** This research received no external funding.

**Institutional Review Board Statement:** Not applicable.

**Informed Consent Statement:** Not applicable.

**Data Availability Statement:** The data presented in this study are available on request from the authors.

**Acknowledgments:** We would like to express our special thanks to the Department of Plant Production and Genetics, School of Agriculture, Shiraz University, for the initial financial support given to conduct this study.

**Conflicts of Interest:** The authors declare no conflict of interest.

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
