# Peer review of "Nano-Organic Coatings Improve Early Vigor of Brassica napus L. Seeds in Water Deficit"

_agronomy, doi:10.3390/agronomy13020390_

Round 1

Reviewer 1 Report

This is an original research article, which highpoints the effects of some organic coatings on the vigor of Canola seeds under water deficit.

The topic presents interest because canola seeds are very intensely cultivated around the world.

The article can be accepted for publication with minor revisions:

  1. It would be necessary to mention the years in which these two experiments were conducted.
  2. The article is very good, and the introduction part provides sufficient information about canola, about nanoparticles. There is a lack of information about drought stress thought. Some mistyping are also present (line 37, 38)
  3. In the Materials and Methods part, the authors should better describe the method used for obtaining the nanoparticles or give a citation for this.
  4. In the Materials and Methods parts there is described that the authors used 3 concentrations of nanoparticles (0mM, 0.5 mM, and 1 mM), but in the Results part is not very clear which concentration it was used.
  5. In the Discussion part, it would be necessary to correlate the obtained results with other recently published studies.
  6. I would recommend making a figure with the work plan of the experiments to better understand the results obtained.

Author Response

REVIEWER#1

This is an original research article, which highpoints the effects of some organic coatings on the vigor of Canola seeds under water deficit.

The topic presents interest because canola seeds are very intensely cultivated around the world.

 The article can be accepted for publication with minor revisions:

  1. It would be necessary to mention the years in which these two experiments were conducted.

AUTHORS: Thanks to the notice of the honorable referee, this test was carried out in 2018-2019, which was added in the materials and methods section as follows.

Experiment 1 was performed in 2018 to find the best combination of these treatments to be used as coating for seeds to help seeds germinate more properly and in turn help seeds to emerge and absorb the moisture………..

After the screening the treatments using the first experiment, Experiment 2 was conducted in 2019 in a greenhouse located in College of Agriculture, Shiraz University, as a factorial experiment arranged in a completely randomized design with five replications.

  1. The article is very good, and the introduction part provides sufficient information about canola, about nanoparticles. There is a lack of information about drought stress thought. Some mistyping are also present (line 37, 38)

AUTHORS: The comment of Reviewer is pertinent. We have used the word water stress instead of drought in the text of the article, especially in the introduction, and in lines 48 to 60 I have talked about its effect on the germination and establishment of canola seedlings. See below the sentence highlighted in yellow. We remain available to integrate and to argue better if you deem it necessary.

On the other hand, plants are reportedly more sensitive to water stress at the germination stage. Drought stress in this stage reduces the longitudinal growth of the shoot and root and delays the time of emergence (Yazdani et al, 2018) and reduces the uniformity seedling emergence and density in the field. In this regard, coating method for pretreating seeds and improving their traits can bring about great advantages under this situation [1]. The absorption of water which depends both on water storage of the seed bed and seed characteristics is the most essential factor for hydration in the early stages of seed germination. Water shortage in the seed bed leads to osmotic stress in seeds potentially causing negative effects and preventing the seed from proper germination [5].

Errors in lines 37 and 38 were corrected in the following way:

Other researchers claimed that pretreatment of seeds can also improve the early vigor of seeds [4]. Different pre-treatments of seeds with either organic or inorganic compounds have been implemented on different conditions and plant species. Seed coating or seed pelleting is an important part of pretreatment. In this method, changes in the shape, dry weight, size, or seed surface structure occur using different amounts of materials and compounds [1]. 

  1. In the Materials and Methods part, the authors should better describe the method used for obtaining the nanoparticles or give a citation for this.

AUTHORS: Thanks for the good comment, referee I have added the following to the material and methods section (which is highlighted in yellow) about nanoparticles. Nanoparticles were purchased from Nano Pars Lima Co, their characteristics are listed in Table 1

  1. In the Materials and Methods parts there is described that the authors used 3 concentrations of nanoparticles (0mM, 0.5 mM, and 1 mM), but in the Results part is not very clear which concentration it was used.

AUTHORS: The effect of nano application on germination and emergence traits of seedling traits is shown in Tables 3, 4, 6 and 7 and its effect on enzymes under drought stress conditions is shown in Tables 8, 10 and 11 and is explained from line 183 to 310 in the text and appropriate Its highest level is also described. For example in line 230- 237:

The emergence rate of canola seeds increased from 0.17and 0.25(seed/day) by applying the nanoparticles under 100% FC, and only from 0.11% to 0.17% under 50% FC (Table 6). Similarly, seedling vigor index increased from 248 under control conditions under no water stress to 352 by applying the nanoparticles under 100% FC, while these ratios under 50% FC were only from 175 and 255, respectively (Table 6).

or in line 230- 237:

The lowest ratios for superoxide dismutase (34), peroxidase (23.49), catalase (3.71), chlorophyll content (48.60) and the highest of electrolyte leakage (50.04) were obtained at the control with no application of this nanomaterial, while their highest rates of superoxide dismutase, peroxidase, catalase, chlorophyll content and lowest of electrolyte leakage were observed in highest level of nano-titanium (1 mM) (Table 8)

If the reviewer wants more details, I remain available to make further additions.

  1. In the Discussion part, it would be necessary to correlate the obtained results with other recently published studies.

AUTHORS: The reviewer's observation is correct. The Discussion has been revised and more updated references have been added to improve the interpretation of the results. We remain available for further additions and improvements.

  1. I would recommend making a figure with the work plan of the experiments to better understand the results obtained.

AUTHORS: The outline of the applied experimental protocol has been added to the manuscript as Figure 1

Reviewer 2 Report

I have reviewed the article entitled “Nano- organic coatings improve early vigor of Canola (Brassica  napus L.) Seeds by underwater deficit”

. The research study is very interesting and meaningful, this study was to find the best combination of nano-organic be used as  coating for seeds for strengthening the absorption of water and emergence seed of low vigor canola.  The treatments in the first experiment  included six levels of organic coatings: 20% vermicompost  with 80% bentonite, 30% vermicompost with 70% bentonite, 40% vermicompost with 60% bentonite,  40% residues of canola with 60% bentonite, 30% residues of canola with 70% bentonite, and 20%  resides of canola with 80% bentonite. The results indicated that 20% residues of canola and 20% vermicompost had the highest moisture under the saturation conditions and kept moisture for a longer time. Subsequently, the factors of the second experiment included three levels of water stress  (100, 75 and 50% field capacity (FC)),  three levels of seed coating (control, vermicompost, and canola  residues), and three levels of titanium dioxide nanoparticles (0, 0.5 and 1 mM). The results showed  that nano-organic coating improved emergence percentage and seedling traits while increasing su- peroxide dismutase, peroxidase, catalase activities and dicreasing electrolyte leakage. The results of  this study suggested nano-canola residue coating to improve the seed emergence traits of canola in  seed broadcasting under water stress conditions.

However, the captions in Tables and Figures should be amended.
In addition, English is decent but I suggest a thorough review of the manuscript before accepting it for publication.

To further improve the text, I suggest the following changes in the manuscript.

Please pay attention on the use of full stop and commas

Abstract
1.      Make the title a simple statement.
2.      Give the problem statement in a single line.
3.      Give a reason for the selection of the current technique.
4.      Quantitative data is also important to support your conclusion. Would you please provide some quantitative data in terms of percentage significant increase or decrease in the abstract?
5.      Please provide a conclusive conclusion with is withdrawn through research in a single line.
6.      Give future prospective in a single line.
7.      As per standard suggestions, please avoid using title words as keywords.

Line 11; add as a coating

Line 14; with 70% bentonite, 40% vermicompost with 60% bentonite,  40%?

What do your mean of  bentonite,  40%? Is it twise?

Line 23; remove the before seed emergence

8.  Please follow the title and improve the introduction in the following sequence as i.e., rubber tree genotypes in the early tapping phase in the Colombian Amazon, problem statement, aims of study, and hypothesis.
9.      Also, provide a novelty statement at the end. What new things authors have done or correlated in this research compared to old ones?
10.     Would you please give a single line about the knowledge gap which your research has covered along with the hypothesis statement?

Line 29 ;  change to of plant production.

Line 30 change to and quantity of final plant production.

Line 30; Seed has changed to seeds have

Line 34; moisture change to humidity etc.

Line 38 pre-treatment of seeds change to Seed pre-treatment

Line 39-40; seeds can also improve early vigor of seeds. What do you mean please rewrite this whole sentence it’s uncleared

Line 43. shape,  dry  weight,  size of what? You mean seed?

Line 55; Water shortage change to lack of water

Line 57 increase change to improve

Line 60 different  proportions change to varying levels

the aim of this study was to  monitor different types of organic and titanium  dioxide nanoparticles coating of canola  seeds  under  water  stress  conditions  to  determine  the  best  combination  to  enhance  the  early vigor of seeds

You objective is not cleared please rewrite this

In Statistical analysis please gives the reference

Material and methods:
Table 1. Soil properties of experimental pots. Is not in format please correct that .
However, the captions in Tables should be amended
Results and Discussion.
12.     Very descriptive. Please give only significant results. Also, give mechanistic discussion. It is not a correct way to discuss results based on other scientists' findings. Please elaborate on specified mechanisms which are regulating and result

Conclusion
13.     Add the targeted beneficiary audience who will get benefits from this research.
Also, give clear-cut recommendations

14 In spite this is research article a lack of recent literature (Recent references (last 3 years), therefore the authors should include the most recent references on this subject

15. References

Standardize references

Author Response

REVIEWER#2

I have reviewed the article entitled “Nano- organic coatings improve early vigor of Canola (Brassica  napus L.) Seeds by underwater deficit”

The research study is very interesting and meaningful, this study was to find the best combination of nano-organic be used as  coating for seeds for strengthening the absorption of water and emergence seed of low vigor canola.  The treatments in the first experiment  included six levels of organic coatings: 20% vermicompost  with 80% bentonite, 30% vermicompost with 70% bentonite, 40% vermicompost with 60% bentonite,  40% residues of canola with 60% bentonite, 30% residues of canola with 70% bentonite, and 20%  resides of canola with 80% bentonite. The results indicated that 20% residues of canola and 20% vermicompost had the highest moisture under the saturation conditions and kept moisture for a longer time. Subsequently, the factors of the second experiment included three levels of water stress  (100, 75 and 50% field capacity (FC)),  three levels of seed coating (control, vermicompost, and canola  residues), and three levels of titanium dioxide nanoparticles (0, 0.5 and 1 mM). The results showed  that nano-organic coating improved emergence percentage and seedling traits while increasing su- peroxide dismutase, peroxidase, catalase activities and dicreasing electrolyte leakage. The results of  this study suggested nano-canola residue coating to improve the seed emergence traits of canola in  seed broadcasting under water stress conditions.

 However, the captions in Tables and Figures should be amended.
In addition, English is decent but I suggest a thorough review of the manuscript before accepting it for publication.

To further improve the text, I suggest the following changes in the manuscript.

AUTHORS: We thank the reviewer for his helpful suggestions. We have tried to improve the manuscript, further linguistic corrections will be performed together with the editorial office.

 Please pay attention on the use of full stop and commas

Abstract
1.      Make the title a simple statement.

AUTHORS: According to the Reviewer we tried to change the title: "Nano-organic coatings improve early vigor of Brassica napus L. seeds in water deficit". We remain available for further suggestions.

  1.     Give the problem statement in a single line.

AUTHORS: Thanks to the referee's good suggestion, the following sentence was added to the beginning of the abstract

Low emergence and vigor of canola seed are common problems encountered by canola growers

  1.     Give a reason for the selection of the current technique.

AUTHORS: The following short sentence was added to the abstract for the reason for using this technique

Seed coating is a good way to solve this problem

  1.     Quantitative data is also important to support your conclusion. Would you please provide some quantitative data in terms of percentage significant increase or decrease in the abstract?

 AUTHORS: For quantitative data the, the following sentence was added to the abstract

The highest ratios for emergence percentage (74.67), seedling vigor index (264.58), and leaf area (418) were obtained at the highest level of nano-titanium (1 mM). The highest rates of superoxide dismutase(53.44), peroxidase(30.68), catalase(5.35), chlorophyll content(51.05), and lowest electrolyte leakage(42.39) were observed in the highest level of nano-titanium (1 mM).

  1.     Please provide a conclusive conclusion with is withdrawn through research in a single line.
    AUTHORS: Based on the referee's opinion, this sentence "The results of this study suggested nano-canola residue coating to improve the seed emergence traits of canola in seed broadcasting under water stress conditions " was replaced with the following sentence

Based on the results, we suggest the use of rapeseed residues with nano titanium oxide for rapeseed coating.

6- Give future prospective in a single line.

AUTHORS: For future prospective the following sentence was added to the  end of the abstract

The use of nanoparticles in current seed coatings is useful for reducing stresses in the germination and emergence stages

  1.     As per standard suggestions, please avoid using title words as keywords.

AUTHORS: The only common word between the keywords and the title was the word canola, which was removed

Line 11; add as a coating

AUTHORS: Was corrected

Line 14; with 70% bentonite, 40% vermicompost with 60% bentonite,  40%?

What do your mean of  bentonite,  40%? Is it twise?

Bentonite is a swelling clay consisting mostly of montmorillonite soil and is a powerful absorbent of water and liquids and has applications in construction and civil affairs, agriculture and horticulture, animal husbandry, etc. But For more clarity, the following sentence was added to the Materials and Methods section

Bentonit is a swelling clay consisting mostly of montmorillonite soil and is a powerful absorbent of water and liquids. Bentonit were purchased from Sinato Co.

Line 23; remove the before seed emergence

AUTHORS: This sentence was replaced by the following two sentences at the request of referee

Please follow the title and improve the introduction in the following sequence as i.e., rubber tree genotypes in the early tapping phase in the Colombian Amazon, problem statement, aims of study, and hypothesis.

AUTHORS: In order to express the importance of the topic and the clarity of the statement of the hypothesis, the following two sentences were added to the introduction

Canola seed is small and sensitive to drought stress during germination and seedling establishment (Taghvaei and cjichi, 2009). Drought stress reduces the rate of daily germination and greatly reduces the number of established seedlings, and this problem occurs when replanting is not possible[Rezayian et al, 2018]. Our hypothesis is that the use of organic coatings with titanium nanoparticles can reduce the drought stress in the germination and emergence stage by absorbing and retaining water around the rapeseed, increasing the rate of emergence and finally the number of established plants.

If more content is needed, let us know

  1.     Also, provide a novelty statement at the end. What new things authors have done or correlated in this research compared to old ones?

AUTHORS: In order to express the novelty  the following sentence were added to the introduction

The novelty of this research is the combined use of organic matter, bentonite, and titanium nano-oxide materials to enhance the germination and emergence of rapeseed in the conditions of drought stress in the germination stage.

  1.    Would you please give a single line about the knowledge gap which your research has covered along with the hypothesis statement?

AUTHORS: In order to express the knowledge gap the following sentence was added to the introduction

There are few reports on the use of nanoparticles in seed coating and almost no reports on the use of nano titanium dioxide in rape seed coating.

Line 29 ;  change to of plant production.

AUTHORS: It was corrected

Line 30 change to and quantity of final plant production.

AUTHORS: It was corrected

Line 30; Seed has changed to seeds have

AUTHORS: It was corrected

Line 34; moisture change to humidity etc.

AUTHORS: It was corrected

Line 38 pre-treatment of seeds change to Seed pre-treatment

AUTHORS: It was corrected

Line 39-40; seeds can also improve early vigor of seeds. What do you mean please rewrite this whole sentence it’s uncleared

AUTHORS: The sentence was modified as follows for clarity

Other researchers claimed that seed pre-treatment can also improve the germination and emergence percentage of seeds

Line 43. shape,  dry  weight,  size of what? You mean seed?

AUTHORS: Thanks to the reviewer's accuracy, the word seed was added for the meaning of the sentence

In this method, changes in the shape, dry weight, size of seed or seed surface structure occur using different amounts of materials and compounds [1].

Line 55; Water shortage change to lack of water

AUTHORS: Thanks to the reviewer's accuracy, the sentence was corrected

Line 57 increase change to improve

AUTHORS: It was corrected

Line 60 different  proportions change to varying levels

AUTHORS: It was corrected

the aim of this study was to  monitor different types of organic and titanium  dioxide nanoparticles coating of canola  seeds  under  water  stress  conditions  to  determine  the  best  combination  to  enhance  the  early vigor of seeds

You objective is not cleared please rewrite this

AUTHORS: Thanks to the reviewer's accuracy, the aim sentence was replaced with the following sentence

The aim of this research is to investigate the effect of using organic materials together with hydrophilic minerals such as bentonite and nano titanium oxide in order to create a hydrophilic compound capable of absorbing and retaining water for a period of time longer in the ground. The combination of these components will create a coating suitable for improving germination and emergence of oilseed rape under conditions of drought stress.

In Statistical analysis please gives the reference

AUTHORS: The reference to software  was added to the Statistical analysis section

2.4 Statistical analysis

Prior to analysis, the data population normality was verified by Kolmogorov–Smirnov with SPSS  (10.0) software (IBM, Armonk, New York, USA). The data were analyzed using ‘SAS’ statistical software v.9.3(SAS Institute Inc.) after ensuring the homogeneity of variances. The means were compared by Duncan’s multiple range (DMRT) test in cases where the F-test of the ANOVA table indicated a significant difference at least at the P < 0.05 and P < 0.01 level and curve was drowned by Excel (Microsoft Inc., Chicago, IL).

Material and methods:
Table 1. Soil properties of experimental pots. Is not in format please correct that .However, the captions in Tables should be amended

AUTHORS: The title of the table was modified as follows

Table 1. Some physical and chemical characteristics of titanium dioxide nanoparticles and soil used in the experiment pots.

Results and Discussion.
12.     Very descriptive. Please give only significant results. Also, give mechanistic discussion. It is not a correct way to discuss results based on other scientists' findings. Please elaborate on specified mechanisms which are regulating and result

AUTHORS: We have provided a full explanation of the results so that nothing is ambiguous because in the previous article that we published in this magazine, two referees asked us to explain the results completely. We have explained all the significant effects and have refrained from explaining the effects that were not significant If the referee requests that we delete some parts, please mention them in the text of the results

Conclusion
13.     Add the targeted beneficiary audience who will get benefits from this research.
Also, give clear-cut recommendations

AUTHORS: At the request of the referee, the final sentence was modified as follows

The final suggestion is that rapeseed seeds should be covered using a combination of organic materials, bentonite and titanium nanoparticles for planting in dry ecosystems so that they to be able to cope with the negative effects of water stress and produce more emergence in these conditions.This method is recommended for commercial canola seed sellers

14 In spite this is research article a lack of recent literature (Recent references (last 3 years), therefore the authors should include the most recent references on this subject

AUTHORS: References have been updated and revised as required.

  1. References

 Standardize references

AUTHORS: the Reviewer's observation is pertinent, the references have been formatted following the MDPI rules.

Reviewer 3 Report

The manuscript presents an alternative coating with organic matter and nanoparticles for the pre-treatment of Canola seeds, for greater moisture absorption during the initial stage of development. The results obtained indicate improvements in the vigor parameters of the seeds and a decrease in water stress, which is of great importance applied to crops that grow in drought conditions.

I suggest that the introduction include the term priming, for, in which a great deal of information on seed invigoration has been developed.

I also suggest that the term hydration in the early stages of seed germination be replaced by imbibition, which is well defined as the initial part of the germination stages.

The introduction and discussion should make some mention of the possible effect of mycorrhizae on the seeds. The foregoing considering the biological products that are evaluated, such as vermicompost and canola residues.

I suggest that the manuscript be carefully reviewed to eliminate formatting and writing errors.

Additionally, I suggest the following changes:

Page 2, line 89, indicate the conditions for obtaining the vermicompost and if you have any proximal analysis of the product.

Page 2, line 89, indicate what type of bentonite used: sodium, calcium, mesosodium.

Page 2, lines 90-91, describe what the canola residue consists of and if you have proximal composition data.

Page 3, line 114, indicate the number of replicates for each treatment, this data is important because significant differences are mentioned in the results.

Page 4, line 166, correct the word absobtion for "absorption" in Figure 1.

Page 6, lines 208-210. Table 3 is very difficult to understand. Not all abbreviations are defined, and an explanation of the calculation procedure is not given. In this same table, very high CV values are observed for % emergence and the Seed Vigor Index, which indicates a low significance of the results. I suggest that a Chi-square test be included in this table to determine the probability of these data.

Page 8, Lines 285-287. The units of the variables described in the table must be indicated correctly. Do the same for tables 8, 9 and 10.

Author Response

REVIEWER#3

The manuscript presents an alternative coating with organic matter and nanoparticles for the pre-treatment of Canola seeds, for greater moisture absorption during the initial stage of development. The results obtained indicate improvements in the vigor parameters of the seeds and a decrease in water stress, which is of great importance applied to crops that grow in drought conditions.

I suggest that the introduction include the term priming, for, in which a great deal of information on seed invigoration has been developed.

AUTHORS: Our research is about seed plating, although like priming, it enhances germination and seedling establishment, but the two categories are separate from each other, but according to the referee's emphasis, we added the following sentence to the introduction to clarify the difference with priming.

Seed priming has been used to improve germination, reduce seedling emergence time, and ameliorate stand establishment and yield [ Adhikari, 2022]

 Adhikari, B.; Olorunwa, O.J.; Barickman, T.C. Seed Priming Enhances Seed Germination and Morphological Traits of Lactuca sativa L. under Salt Stress. Seeds 2022, 1, 74-86, doi:10.3390/seeds1020007.

I also suggest that the term hydration in the early stages of seed germination be replaced by imbibition, which is well defined as the initial part of the germination stages.

 AUTHORS: It was replaced

The introduction and discussion should make some mention of the possible effect of mycorrhizae on the seeds. The foregoing considering the biological products that are evaluated, such as vermicompost and canola residues.

AUTHORS: In this research, we have studied the effect of organic materials to obtain the most suitable coating, and we have not discussed the effect of microbes, which is far from our topic.

I suggest that the manuscript be carefully reviewed to eliminate formatting and writing errors.

AUTHORS: the reviewer's observation is pertinent. We have attempted to correct all errors, however further verification will be carried out with the editorial office.

Additionally, I suggest the following changes:

Page 2, line 89, indicate the conditions for obtaining the vermicompost and if you have any proximal analysis of the product.

AUTHORS: The table 1 was add to material and methods section.

Page 2, line 89, indicate what type of bentonite used: sodium, calcium, mesosodium.

AUTHORS: Thanks to the accuracy of the reviewer, bentonite type was added to the materials and methods section

Bentonit)Sodium type)is a swelling clay consisting mostly of montmorillonite soil and is a powerful absorbent of water and liquids. Bentonit were purchased from Sinato Co.

Page 2, lines 90-91, describe what the canola residue consists of and if you have proximal composition data.

The proximal composition of canola added to table 1

Page 3, line 114, indicate the number of replicates for each treatment, this data is important because significant differences are mentioned in the results.

AUTHORS: Thanks to the accuracy of the reviewer, In line 104, in the sentence below, we have stated the number of repetitions

Experiment 2 was conducted in a greenhouse located in College of Agriculture, Shiraz University, as a factorial experiment arranged in a completely randomized design with five replications

Page 4, line 166, correct the word absobtion for "absorption" in Figure 1.

AUTHORS: It was corrected

Page 6, lines 208-210. Table 3 is very difficult to understand. Not all abbreviations are defined, and an explanation of the calculation procedure is not given. In this same table, very high CV values are observed for % emergence and the Seed Vigor Index, which indicates a low significance of the results. I suggest that a Chi-square test be included in this table to determine the probability of these data.

AUTHORS: Abbreviations are only in the first column from the left, and I added their explanation below the table as following

           **, * and ns: significant at the 1% , 5% probability level and not significant, W= Moisture regime, C= Coating, N=Nano particle,

About of Table 3

Table 3 is the results of the variance analysis of the test data, which was calculated in the form of a completely random design in 5 repetitions, which we have described in the materials and method as following

About of an explanation of the calculation procedure

 Experiment 2 was conducted in 2019 in a greenhouse located in the College of Agriculture, Shiraz University, as a factorial experiment arranged in a completely randomized design with five replications.

 About of coefficient of variation

 But regarding the coefficient of variation, you should note that this experiment was conducted in the greenhouse and in 5 replicates, and we had a lot of fluctuations in one of our replicates, although we could remove this replicate and then analyze the data with 4 replicates.

But we preferred to analyze the data with the same 5 repetitions, as a result of which our coefficient of variation showed some increase. Because the vigor index is also obtained by multiplying the percentage of emergence with the length of the seedling, therefore, the increase in the coefficient of variation in the index of weighted root has also increased. On the other hand, the coefficient of variation for greenhouse and then field tests is more than the tests performed in the growth chamber and its value is acceptable up to 30%.

About of chi-square

But as for chi-square, this index is mostly used to evaluate ranked qualitative data, but our data is quantitative, and its use is not recommended for quantitative data, although I would like to know your further recommendations on this matter.

Page 8, Lines 285-287. The units of the variables described in the table must be indicated correctly. Do the same for tables 8, 9 and 10.

AUTHORS: It has been corrected. However, if there is anything else that needs to be done to correct, please let me know.

Round 2

Reviewer 2 Report

Author made all changes in the manuscript now it should be accepted.